# The Role of Left Atrial Longitudinal Strain in the Diagnosis of Acute Cellular Rejection in Heart Transplant Recipients

**DOI:** 10.3390/jcm11174987

**Published:** 2022-08-25

**Authors:** Sara Rodríguez-Diego, Martín Ruiz-Ortiz, Mónica Delgado-Ortega, Jiwon Kim, Jonathan W. Weinsaft, José J. Sánchez-Fernández, Rosa Ortega-Salas, Lucía Carnero-Montoro, Francisco Carrasco-Ávalos, José López-Aguilera, Amador López-Granados, José M. Arizón del Prado, Elías Romo-Peñas, Laura Pardo-González, Francisco J. Hidalgo-Lesmes, Manuel Pan Álvarez-Ossorio, Dolores Mesa-Rubio

**Affiliations:** 1Cardiology Department, Reina Sofia University Hospital, 14004 Cordoba, Spain; 2Greenberg Division of Cardiology, Weill Cornell Medical College, New York, NY 10065, USA; 3Pathology Department, Reina Sofia University Hospital, 14004 Cordoba, Spain

**Keywords:** rejection, left atrial strain, speckle-tracking echocardiography, heart transplantation

## Abstract

Our aim was to investigate the role of left atrial longitudinal strain (LALS) in the non-invasive diagnosis of acute cellular rejection (ACR) episodes in heart transplant (HTx) recipients. Methods: We performed successive echocardiographic exams in 18 consecutive adult HTx recipients in their first year after HTx within 3 h of the routine surveillance endomyocardial biopsies (EMB) in a single center. LALS parameters were analyzed with two different software. We investigated LALS association with ACR presence, as well as inter-vendor variability in comparable LALS values. Results: A total of 147 pairs of EMB and echo exams were carried out. Lower values of LALS were significantly associated with any grade of ACR presence. Peak atrial longitudinal strain (PALS) offered the best diagnostic value for any grade of ACR, with a C statistic of 0.77 using one software (95% CI 0.68–0.84, *p* < 0.0005) and 0.64 with the other (95% CI 0.54–0.73, *p* = 0.013) (*p* = 0.02 for comparison between both curves). Reproducibility between comparable LALS parameters was poor (intraclass correlation coefficients were 0.60 for PALS, 95% CI 0.42–0.73, *p* < 0.0005; and 0.42 for PALS rate, 95% CI −0.13–0.68, *p* < 0.0005). Conclusions: LALS variables might be a sensitive marker of ACR in HTx recipients, principally discriminating between those studies without rejection and those with any grade of ACR. Inter-vendor variability was significant.

## 1. Introduction

Left atrial longitudinal strain (LALS) has recently emerged as a useful tool, generating independent added value in different scenarios such as valvulopathies, atrial fibrillation, heart failure and systemic diseases [1,2,3,4,5,6,7,8,9,10]. It has also been postulated as a means for evaluating diastolic function and left ventricular filling pressures [11,12,13,14], reported to be altered in acute cellular rejection (ACR) episodes in heart transplant (HTx) recipients [15,16].

Endomyocardial biopsy (EMB) histological analysis is the gold-standard diagnostic method for cardiac graft ACR [17]. In 2005, the International Society for Heart and Lung Transplantation (ISHLT) published a revised consensus classification for ACR grading after HTx based on the inflammatory infiltrate severity [18]. Although it is an invasive and expensive procedure, EMB provides ACR detection and therefore treatment initiation in moderate and severe grades. Moreover, recurrent mild ACR (non-treatment requiring) episodes after HTx have been associated with poorer long-term outcomes [19], emphasizing the prognostic impact of any grade of ACR diagnosis.

As an arising technology, myocardial strain analysis by means of speckle-tracking echocardiography has been tested in ACR. In that regard, left ventricular global longitudinal strain (GLS) has been proved to be more significantly associated with myocardial injury in ACR episodes than the ejection fraction, which is usually preserved until severe damage is established [20]. Some studies also expose left and right ventricular GLS diagnostic value in this setting [21,22,23,24,25,26]. While promising, these results have not yielded a universal cut-off limit for ACR exclusion.

Whereas strain analysis techniques applied to ventricular walls and their association with ACR in HTx recipients have been widely explored, evidence of the role of LALS in this context is lacking. There are some studies that explore the usefulness of left atrial strain in pediatric HTx recipients, and its association to left ventricular filling pressures [27,28].

Our group reported strain analysis applied to ventricular walls and its variability between two different software [29] in adult HTx recipients. Our aim in this study was to investigate in the same sample if ACR episodes affect LALS (measured by means 2D speckle-tracking echocardiography), the diagnostic impact of LALS in ACR episodes after HTx, and its robustness when analyzed with two different software.

## 2. Materials and Methods

### 2.1. Study Population

We prospectively included all consecutive HTx recipients who underwent routine surveillance EMB during their first year after HTx, from 15 September 2014 to 31 October 2016. Bicaval technique was the surgical approach in all patients.

An echocardiographic exam was performed within 3 h of EMB. Baseline demographics, clinical characteristics and hemodynamic data obtained in right heart catheterization were collected for each patient.

ACR episodes were managed according to clinical presentation, EMB results and medical standards by expert heart failure cardiologists who were blinded to strain analysis results. No treatment was initiated before echocardiographic examination had been performed, as EMB histological reports were not yet available.

The study fulfilled Helsinki recommendations for medical studies and was validated by the ethical research board of our center (approval code: “EcocardioHRS-2014-1”). All patients agreed and granted informed consent before enrollment.

### 2.2. Endomyocardial Biopsy

In our center, surveillance protocol after HTx consists of periodical transjugular vein right heart catheterization with EMB and right hemodynamic study (right atrial and ventricular, pulmonary artery and wedge pressures). EMB is performed weekly in the first month, once every two weeks until the third month, monthly up to the sixth month, in the ninth and twelfth months, as well as every time there is a clinical surmise of ACR.

Histological examination of EMB specimens was done by an expert pathologist blinded to echocardiographic findings and classified according to the revised criteria of the ISHLT as OR grade (no rejection), 1R grade (mild rejection), 2R grade (moderate rejection) and 3R grade (severe rejection). Grades 2R and 3R were considered treatment-requiring ACR (TR-ACR).

### 2.3. Echocardiographic Study

Image acquisition was performed in accordance with consensus guidelines [30] by the same expert echocardiographer (M.R.O.) using a Siemens ACUSON SC2000 echocardiographic system and a 4V1c 1.25–4.5 MHz transducer (Siemens AG, Erlangen, Germany). In order to examine LALS, an optimized apical four-chamber view comprising at least three cardiac cycles was recorded (frame rates 61–95), and digitally stored as raw data.

### 2.4. Strain Analysis

Strain analysis was performed or closely supervised off-line by two expert operators (M.R.O and J.K.). Both software packages included a dedicated mode for atrial analysis. The atrial region of interest was manually traced and adjusted to the atrial contour. Software automatically executed longitudinal deformation analysis under visual control of the atrial border tracking by the operator, who remained blinded to EMB results. No segmental sub-division of the LA was made, following current consensus for atrial deformation imaging [31]; therefore, only global and average values were considered. Images with poor definition of atrial border or patent failure of tracking (even after adjustments) were discarded from the analysis. Zero strain reference was settled at end-diastole. The end-systole point was manually set.

The studies were firstly analyzed with Tom Tec 2D Cardiac Performance Analysis software (Tom Tec Imaging Systems, Munich, Germany) in New York (USA). LALS variables obtained as output included three longitudinal strain variables: end-systolic global LALS (ESALSg) which stands for the top atrial deformation value reached at the end of the systole, peak average LALS during reservoir phase (PALS) which is the parameter mostly recommended in consensus papers [31] and stands for the average maximum deformation during the whole cycle (reservoir strain), end-systolic average LALS (ESALS) which is the top average deformation at the end of systole.

The sample was secondly analyzed with Velocity Vector Imaging software (Siemens AG, Erlangen, Germany) in Córdoba (Spain), which only proffered one atrial strain variable: PALS.

Both software provided just one strain rate variable: peak average LALS rate during reservoir phase (PALSR).

In order to test inter-vendor variability only comparable variables were used for this purpose: PALS and PALSR. Figure 1 illustrates the output generated by both software.

### 2.5. Groups of Comparison

All echocardiographic studies were classified according to the rejection grade observed in the concomitant EMB in three groups: no rejection (grade 0R), mild rejection (grade 1R) and TR-ACR (grades 2R and 3R). As the primary endpoint for this study, LALS variables and their association with ACR grades were investigated. We also performed a longitudinal analysis, comparing LALS values in every echocardiographic study with the previous one in the same patient, according to the change in rejection status.

### 2.6. Variability Analysis

All valuable studies were incorporated to assess inter-software variability. To evaluate intra-software variability, a random sample of ten studies was analyzed independently by two expert operators with each software. The same sample was newly analyzed one month later by the same operator to test intra-observer variability within each software.

### 2.7. Statistical Methods

Quantitative variables were expressed as mean ± standard deviation if normal distribution was confirmed by Kolmogorov–Smirnov test, or as median [percentile 25–75] otherwise. Categorical variables were expressed as percentages.

Analysis of variance for independent data, Student *t*-test or Mann–Whitney U test were used to investigate the association LALS variables with ACR grades, as appropriate. To examine discriminative ability to diagnose rejection, receiver-operating characteristic (ROC) curves were depicted for those strain variables significantly associated with rejection.

In order to compare strain values generated with both software, Student *t*-test for paired data was used, calculating Pearson correlation coefficients and the slope of regression line. Intra-class correlation coefficients (ICC), bias and Bland–Altman limits for agreement were calculated for all reproducibility analysis.

ROC curves were compared with a De Long test (MedCalc Software, Mariakerke, Belgium). For the rest of statistical analyses, we used SPSS v 21.0 (SPSS Inc., Chicago, IL, USA). A two-tailed *p* value < 0.05 was considered significant.

## 3. Results

### 3.1. Patient Characteristics and Endomyocardial Biopsies Results

We identified 169 EMB performed in 18 HTx recipients during the period of study. As echocardiographic equipment was unavailable in 22 of them, the final sample comprised 147 pairs of EMB and echocardiographic study (8 ± 3 per patient). The HTx recipients’ mean age was 51 ± 14, and the donors’ was 42 ± 11. Most recipients and donors were men (78% and 67%, respectively). Mean ischemic time was 247 ± 33 min. Previous diagnosis leading to transplant were myocardiopathy (*n* = 8, 44%), coronary artery disease (*n* = 7, 39%), valvular disease (*n* = 2, 11%), and congenital heart disease (*n* = 1, 6%). Treatment at first visit consisted of corticosteroids (*n* = 18, 100%), tacrolimus (*n* = 17, 94%), mofetil mycophenolate (*n* = 14, 78%), everolimus (*n* = 6, 33%) and cyclosporine (*n* = 1, 6%). Hemodynamic data collected during EMB procedure included: systolic blood pressure 149 ± 18 mmHg, diastolic blood pressure 94 ± 12, heart rate 93 ± 11 beats/min, systolic right atrial pressure 4 ± 3 mmHg, systolic right ventricular pressure 35 ± 7 mmHg and end-diastolic right ventricular pressure 5 ± 3 mmHg.

Pathological diagnosis was absence of ACR (grade OR) in 65 EMB (44%), mild rejection (grade 1R) in 63 EMB (43%), and TR-ACR in 19 EMB (13%), without significant differences in ACR grades distribution when compared to the whole sample (*n* = 169, 0R: 42%, 1R: 44% and ≥2R: 14%, *p* = 0.24). Conforming to histological and immunopathological analysis, none of the EMB met criteria for antibody-mediated rejection. Although being asymptomatic at the moment of examination, patients with TR-ACR (19 episodes in 9 patients) received intensified immunosuppressant therapy. There were neither deaths nor hemodynamic instability episodes.

Regarding echocardiographic analysis, conventional and ventricular strain variables in this series were reported at length by our group in a previous paper [29].

### 3.2. Association of LALS and Presence of ACR

LALS analysis was firstly performed with TomTec software. Among 147 echocardiographic exams, LALS parameters were evaluable in 131 (89.1%) with TomTec software, and in 114 (77.5%) with Siemens (*p* = 0.01). We found that lower values of LALS parameters were significantly associated with higher ACR severity (Table 1). Post hoc subgroup analysis and binary comparisons (Table 1 and Table 2) pointed to a discriminative value between ACR absence and presence of any grade of ACR.

LALS parameters performance in the diagnosis of any grade of ACR was calculated depicting ROC curves (Figure 2). On ROC analysis, PALS obtained with TomTec software (TT-PALS) had the highest area under the curve (0.74, 95% CI 0.64–0.82, *p* < 0.0005). A value of TT-PALS ≥ 19%, present in 34% of studies, had a negative predictive value of 77% for any grade of ACR.

Regarding longitudinal analysis, the previous echocardiographic study was available for 129 studies: rejection status changed from ACR absence to ACR presence in 15 cases, it did not change in 91 pairs (36 showed no rejection at the two time points and 51 presented ACR at both of them) and it changed from ACR presence to no rejection in 23 cases. LALS parameters did not significantly change according to changes in rejection status (Figure 3).

### 3.3. Variability of LALS Parameters

Values of comparable LALS parameters (PALS and PALSR) were significantly correlated, although Pearson’s correlation coefficients of 0.44 to 0.45 suggest only a moderate association (*p* < 0.001 for both correlations, Figure 4, lower panels). However, reproducibility was poor: ICC for PALS was 0.60, 95% CI 0.42–0.73, *p* < 0.0005; and 0.42 for PALSR, 95% CI −0.13–0.68, *p* < 0.0005. Bland–Altmann plots (Figure 4, upper panels) showed wide limits of agreement for both parameters, with significant bias for PALSR (*p* < 0.0005).

Intra-observer and inter-observer variability were good to very good with both software, with ICC fluctuating from 0.74 to 0.94 (Table 3).

As PALS was the parameter with the best ability to discriminate the presence of any grade of ACR, ROC curves were drawn using a total of 106 studies in which PALS values obtained with both software were available (Figure 5). Both software showed a discriminative ability for any grade of ACR diagnosis (AUC was 0.77 with TomTec, 95% CI 0.68–0.84, *p* < 0.0005; and 0.64 with Siemens, 95% CI 0.54–0.73, *p* = 0.013). The difference observed between areas was significant (*p* = 0.02), demonstrating TomTec’s better discriminative ability.

### 3.4. Sensitivity and Specificity of Atrial Strain Parameters for Diagnosis of Rejection

We randomly divided the whole sample in which PALS measurement was feasible with TomTec (*n* = 129) into derivation (*n* = 57) and validation (*n* = 72) cohorts. The ROC curve for PALS was calculated in the derivation cohort, showing a C statistic of 0.67 (95% CI 0.51–0.82, *p* = 0.029). A PALS value of ≥16% was selected as the best cut-off point for any grade ACR diagnosis in the derivation cohort. Applying this cut-off point to the validation cohort, we obtained sensibility, specificity, positive and negative predictive values for the diagnosis of any grade of ACR of 76%, 79%, 81% and 74%, respectively.

## 4. Discussion

The main finding in this study was the significant association of LALS parameters with the presence of any grade of ACR in HTx recipients. To the best of our knowledge, LALS parameters behavior during ACR episodes in adult HTx recipients had not been reported before. LALS values, particularly PALS, showed a significant decrease with the presence of any grade of ACR, but failed to discriminate mild from TR-ACR episodes, and were not able to detect longitudinal changes in rejection status.

These data come from a sample which is similar to those reported in previous studies regarding basal characteristics, as well as ACR episodes incidence [21,22,23,24,25,26] and imaging acquisition was performed according to the recommended frame rates range for atrial strain analysis [31].

A transverse study has shown atrial strain to be affected in HTx recipients compared to control subjects [32], as atrial structure is directly affected by surgical techniques in HTx. However, the association of LALS parameters with rejection found in this study had not been previously reported in adult patients, and we hypothesize it could be related to subtle diastolic dysfunction in the context of ACR episodes. ACR episodes affect myocardium via a different grade of inflammatory infiltrate that damages myocardial function [18]. In this context, LALS might be a sensitive marker of preclinical insult, translating diastolic impairment and a rise in filling pressures [33,34], as shown in children [27,28]. Atrial strain has emerged as a sensitive tool in early detection of subtle or early changes in the setting of various cardiac conditions such as atrial fibrillation, valvulopathies, systemic diseases, heart failure with preserved ejection fraction and diastolic dysfunction [1,2,3,4,5,6,7,8,9,10]. LALS changes usually precede other structural variations such as left atrial volume increase and systolic dysfunction in heart failure. It appears to have a correlation with filling pressures measured by invasive methods, and therefore it might be a very sensitive but less specific tool, a common red flag to multiple heart diseases at subclinical stages. Yeh et al. described PALS and left atrial distensibility potential to predict pulmonary capillary wedge pressure in pediatric HTx recipients [27]. Loar et al., confirmed this finding and proposed that using PALS may be a method of noninvasive monitoring for rejection or nonspecific graft dysfunction in this population [28]. Diastolic strain measurements have also been found useful to identify pediatric HTx recipients at risk of graft coronary artery vasculopathy [35].

However, LALS parameters in our study have been unable to rule out TR-ACR episodes, neither were they able to detect longitudinal changes in rejection status. Bidimensional speckle tracking applied to ventricular walls has shown consistent results in detection of TR-ACR, although this technique has not been as reproducible in longitudinal analysis [21,22,23,24,25,26]. Consistently to this concept of global left ventricular dysfunction affecting both diastolic and systolic performance, our group has recently confirmed the association of the sum of lateral mitral annulus systolic (s’) and early diastolic (e’) velocities (in absolute values) measured by tissue Doppler echocardiography with TR-ACR, a parameter recognized by the current guidelines in the use of imaging in HTx recipients [36]. Moreover, s’ + e’ showed a significant change related to a modification in ACR status in longitudinal analysis. In our study, the low sensitivity of the current technology for LALS measurement or the lack of statistical power could explain the inability of this technique for discriminating TR-ACR or detecting longitudinal changes in rejection status.

Despite well-known strain reproducibility when using the same software, inter-vendor strain reproducibility has been a striking concern during the last decade, due to which the European Association of Cardiovascular Imaging and the American Society of Echocardiography have addressed manifest efforts to standardize measurements. This is also true regarding LALS, and as exposed in the NORRE study, cautions have to be made when using different vendors [37]. As expected, we have found poor reproducibility between strain values obtained by the two software used in this study. However, consistent results regarding ACR detection were obtained. This finding underlines the robustness of the technique for this purpose. Differences in feasibility of strain measurement and discriminative ability between them highlight the importance of software and inter-vendor validation when studying specific populations and pathologies, as recently shown by one of our group’s studies [29].

## 5. Limitations

Our study sample, although representative of standard HTx recipients’ follow-up, accounts for a limited number of patients and is circumscribed to a single center. As an observational study, it can only be hypothesis generating. A multicenter study would help to confirm these findings and may improve statistical power to detect changes in LALS parameters. HTx recipients often present challenging acoustic windows, mainly due to post-surgical changes, and suboptimal quality images forced us to exclude some images from strain analysis. Although our frame rate was optimized and adequate for longitudinal strain analysis, a higher frame rate would have been desirable for strain rate performance. We only took into account four-chamber view for atrial strain analysis, and used PALS during reservoir phase, as this approach is admitted in guidelines and has been used in several studies [5,13,31]. Nonetheless, we acknowledge that it might have been useful to include two-chamber and even three-chamber views to improve global atrial function study and look into differences in reservoir and conduct phases. Nowadays, 3D strain could add diagnostic potency, as it achieves to integrate atrial walls as in a full-volume model. Although cardiac magnetic resonance would have strengthened our findings because of higher spatial resolution and tissue characterization, there was not availability in our center at the moment the study was carried out. Pulmonary wedge pressure was not usually measured in the EMB procedure, so the association of decreased PALS with higher filling pressures could not be confirmed in our study. Whilst we acknowledge that direct infiltration of inflammatory cells in the left atrium could be itself a cause of decreased LALS, there is a scantness of information about atrial histology during acute rejection episodes, since most reports refer to endomyocardial biopsy (which is sampled from right ventricle in surveillance biopsies) or to ventricular histopathology in ex vivo. As LALS could also be affected by graft coronary artery vasculopathy, it would have been desirable to know its prevalence in our sample. This was not available, as we do not perform coronary angiography on a routine basis before the first year of follow-up. While it is true that in classical series graft coronary artery vasculopathy is not as prevalent in the first year after HTx as ACR episodes, the increasing age of donors could lead to a more prevalent coronary artery vasculopathy in current heart grafts. Finally, it would be desirable to know the possible independent prognostic value of LALS in long-term survival after HTx.

## 6. Conclusions

LALS variables were found to be a sensitive marker of ACR episodes in HTx recipients in this prospective observational monocentric study, although more studies with higher sample size and follow-up are required to confirm its potential role in this scenario. The main discriminative value appears to reside between the absence of rejection and presence of any grade of ACR, and these findings were consistent when the images were analyzed using two independent software. Despite high inter-observer and intra-observer agreement, inter-vendor reproducibility of strain values was deficient, and feasibility and discriminative ability were significantly different. We postulate that a LALS decrease could be related to diastolic dysfunction in the context of myocardial impairment.

## Figures and Tables

**Figure 1 jcm-11-04987-f001:**
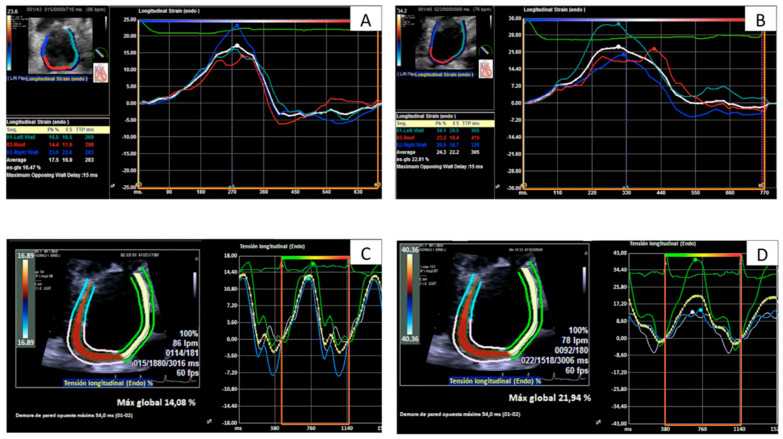
Curves and measurements of peak left atrial longitudinal strain in the same patient with rejection grade 2R (**A**,**C**) and without rejection (**B**,**D**) evaluated by TomTec software (**A**,**B**) and by Siemens software (**C**,**D**).

**Figure 2 jcm-11-04987-f002:**
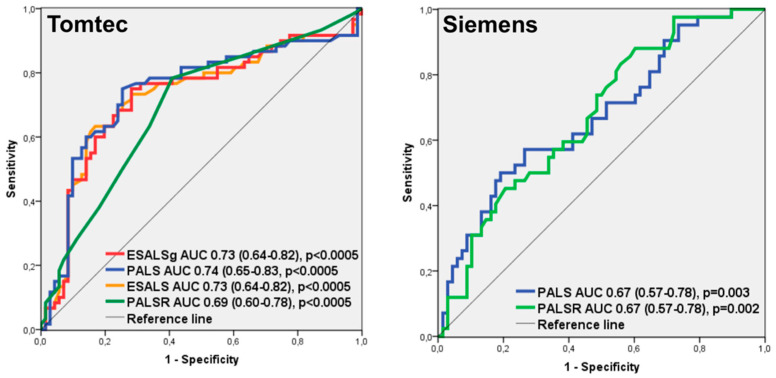
Receiving operator characteristics curves analysis: left atrial longitudinal strain parameters performance in the detection of any grade of rejection. AUC, area under curve; ESLALSg, end-systolic global atrial longitudinal strain; PALS, peak atrial longitudinal strain; ESALS, end-systolic atrial longitudinal strain; PALSR, peak atrial longitudinal strain rate.

**Figure 3 jcm-11-04987-f003:**
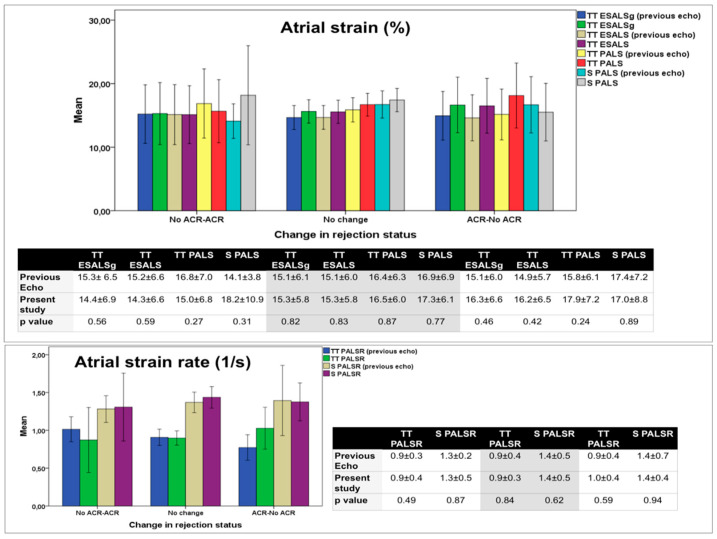
Longitudinal changes in selected variables from the previous echocardiogram to the present study and its association with changes in rejection status. ESALSg, end-systolic global atrial longitudinal strain; PALS, peak atrial longitudinal strain; ESALS, end-systolic atrial longitudinal strain; PALSR peak atrial longitudinal strain rate; TT, TomTec; S, Siemens.

**Figure 4 jcm-11-04987-f004:**
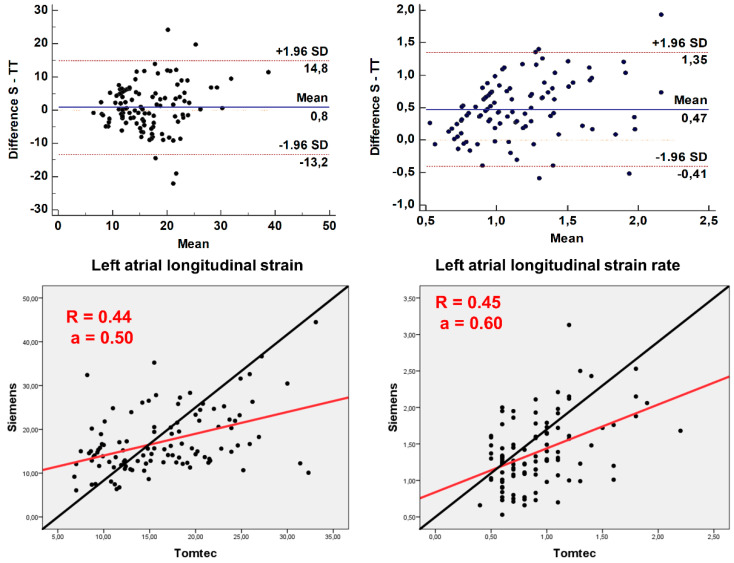
Bland–Altman plots (upper panels) for inter-vendor variability of left atrial longitudinal strain values. Scatter plots (lower panels) for values of strain measured by Siemens (*x*-axis) and TomTec (*y*-axis) software. The Pearson correlation coefficient (R2) and the slope of the regression line (a) are displayed in red. The line of identity is shown in black.

**Figure 5 jcm-11-04987-f005:**
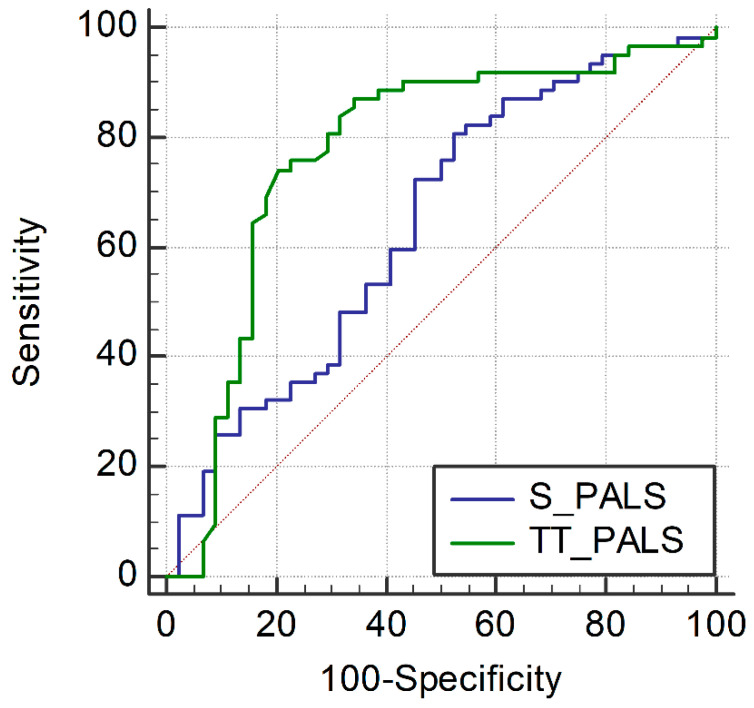
Peak atrial longitudinal strain performance in the diagnosis of any grade of rejection with each software: ROC curves. PALS, peak atrial longitudinal strain; S, Siemens; TT, TomTec.

**Table 1 jcm-11-04987-t001:** Association between left atrial longitudinal strain parameters and different grades of rejection: three-group comparisons.

	Grade of Rejection (ISHLT 2005)			*p* Value			
**Software-Variable**	0R	1R	2R–3R	Overall *	0R vs. 1R **	0R vs. 2R–3R **	1R vs. 2R–3R **
**TT-ESALSg (%)**	17.5 ± 6.0	13.0 ± 5.2	13.5 ± 5.4	**<0.0005**	**<0.0005**	**0.037**	0.94
**TT-PALS (%)**	19.1 ± 6.2	14.2 ± 5.4	13.9 ± 5.1	**<0.0005**	**<0.0005**	**0.006**	0.98
**TT-ESALS (%)**	17.5 ± 6.0	13.0 ± 5.1	13.3 ± 5.4	**<0.0005**	**<0.0005**	**0.025**	0.98
**TT-PALSR (1/s)**	1.0 ± 0.4	0.8 ± 0.3	0.7 ± 0.2	**0.001**	**0.004**	**0.012**	0.70
**S-PALS (%)**	19.4 ± 7.5	15.9 ± 6.4	14.6 ± 6.2	**0.013**	**0.040**	**0.031**	0.75
**S-PALSR (1/s)**	1.5 ± 0.4	1.3 ± 0.5	1.3 ± 0.4	**0.018**	**0.018**	0.15	0.97

* Analysis of variance; ** Post hoc subgroups analysis. ESALSg, end-systolic global atrial longitudinal strain; PALS, peak atrial longitudinal strain; ESALS, end-systolic atrial longitudinal strain; PALSR peak atrial longitudinal strain rate; TT, TomTec; S, Siemens.

**Table 2 jcm-11-04987-t002:** Association between left atrial longitudinal strain parameters and different grades of rejection: binary comparisons.

Software-Variable	0R	Any Grade Rejection	*p* ^a^	0R–1R	TR-ACR	*p* ^b^
**TT-ESALSg (%)**	17.5 ± 6.0	13.1 ± 5.2	**<0.0005**	15.3 ± 6.1	13.5 ± 5.4	0.27
**TT-PALS (%)**	19.1 ± 6.2	14.1 ± 5.4	**<0.0005**	16.7 ± 6.3	13.9 ± 5.7	0.10
**TT-ESALS (%)**	17.5 ± 6.0	13.1 ± 5.1	**<0.0005**	15.3 ± 6.0	13.3 ± 5.4	0.21
**TT-PALSR (1/s)**	1.0 ± 0.4	0.7 [0.6–1.0]	**<0.0005**	0.8 [0.6–1.1]	0.7 ± 0.2	0.053
**S-PALS (%)**	19.4 ± 7.4	14.4 [11.6–7.2]	**0.006**	15.6 [12.5–21.9]	14.6 ± 6.2	0.07
**S-PALSR (1/s)**	1.5 ± 0.4	1.3 ± 0.5	**0.005**	1.4 ± 0.5	1.3 ± 0.4	0.44

*p* ^a^: *p* value obtained by means of Student *t*-test or Mann–Whitney U test, as appropriate, for comparison among echocardiographic studies coincident with any grade of rejection and without rejection. *p* ^b^: *p* value obtained by means of Student *t*-test or Mann–Whitney U test, as appropriate, for comparison among echocardiographic studies coincident with TR-ACR and the rest of studies. TR-ACR: treatment-requiring acute cellular rejection. ESALSg, end-systolic global atrial longitudinal strain; PALS, peak atrial longitudinal strain; ESALS, end-systolic atrial longitudinal strain; PALSR peak atrial longitudinal strain rate; TT, TomTec; S, Siemens.

**Table 3 jcm-11-04987-t003:** Intra-observer and inter-observer variability for left atrial longitudinal strain measurements with Siemens and TomTec software.

Software	Variability	Parameter	ICC (95% CI)	*p*-Value	Bias (%)	Bland–Altman Limits of Agreement (%)
Siemens	Intra-observer	PALS	0.94 (0.77 to 0.98)	<0.0005	1.76	−4.42 to 7.95
		PALSR	0.84 (0.47 to 0.95)	0.003	−0.13	−1.14 to 0.89
	Inter-observer	PALS	0.80 (0.30 to 0.94)	0.007	0.70	−12.16 to 13.56
		PALSR	0.77 (0.17 to 0.94)	0.01	0.01	−0.97 to 0.99
TomTec	Intra-observer	PALS	0.88 (0.50 to 0.97)	0.003	−0.23	−8.49 to 8.03
		PALSR	0.89 (0.58 to 0.97)	0.002	−0.06	−0.46 to 0.34
	Inter-observer	PALS	0.87 (0.51 to 0.97)	0.003	1.58	−7.26 to 10.42
		PALSR	0.74 (−0.14 to 0.94)	0.04	−0.01	−0.59 to 0.57

ICC, intraclass correlation coefficient; CI, confidence interval; PALS, peak atrial longitudinal strain; PALSR, peak atrial longitudinal strain rate.

## Data Availability

Not applicable.

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
