# Peer review of "The Role of Left Atrial Longitudinal Strain in the Diagnosis of Acute Cellular Rejection in Heart Transplant Recipients"

_jcm, 2022, doi:10.3390/jcm11174987_

Round 1

Reviewer 1 Report

Dr. Rodriguez-Diego and colleagues present a prospective analysis involving 18 patients who underwent HTx and examining the association of LALS and ACR. The analysis reveals significant association between LALS and ACR with significant inter-vendor (Tom Tec vs. Siemens) variability. However, there are certain concerns about the hypothesis, design and inferences that require clarification.

Major comments

The main hypothesis regarding the use of LALS is not clear. It should be clearly stated why it would theoretically be superior to ventricular GLS or other echocardiographic markers of ventricular filling pressure or early diastolic dysfunction such as e/e’ or e’.

In contrast to the ventricles, left atrium is extensively sutured during HTx and is inevitably subject to structural changes (possibly adhesion and fibrosis) that would affect LALS in various degrees (depending on the surgical technique and patient’s wound healing properties), which would act as a major confounder. Thus, this reviewer thinks that LALS is not a suitable parameter, especially in HTx patients. Please, elaborate.

Specific comments

Title:

‘strain’ is missing after longitudinal.

Abstract:

Lower value of LALS were associated with higher ACR severity: According to the author’s analysis, LALS could not TR-ACR. Please, explain.

The authors should tone-down about the LALS being a sensitive marker of ACR, considering the small number of sample size, poor reproducibility between the LALS parameters, and inter-vendor variability.

Introduction

A clear background for the hypothesis using the LALS is missing. As mentioned above, the authors should describe what the potential advantage of LALS over ventricular GLS is.

Materials and Methods

Since it is a prospective study, was it registered to an open clinical trial registry? If so, the registry site and number should be provided.

The sample size was newly anlayzed one month later by the same operator to test intraobserver variability within each software: Was the sample size calculated according to this? It seems that the primary endpoint was to assess the association between LALS and ACR. Thus, the sample size should be calculated upon the primary endpoint. Please, explain. Also, the primary and the secondary endpoints need to be clarified in the Methods section.

Results

Unavailable in 22 patients: Reasons should be given as this is a prospective study.

Overall, there are no clear descriptions regarding the unavailable data sets.

TT-PALS 19%, present in 34% of studies: What do the authors mean by that?

Also, please provide both sensitivity and specificity of the cut-off value obtained from ROC analysis.

Discussion

Again, it should be stated why LALS parameters would theoretically be superior to ventricular GLS or other echocardiographic markers of ventricular filling pressure or early diastolic dysfunction such as e/e’ or e’.

Conclusions

Inter-vendor reproducibility of strain values was deficient à present

The authors should tone-down about the LALS being a sensitive marker of ACR, considering the small number of sample size, poor reproducibility between the LALS parameters, inter-vendor variability, and the relatively low AUROC.

Reviewer 2 Report

Dear Authors, 

Please kindly refer to the following comments:

1. Please provide the version of Tomtec Software and Velocity Vector Imaging software.

2. Please describe precisely end-systolic global LALS (and the construction of abbreviation) and end-systolic global average LALS as these parameters are not routinely used and difficult to find in the literature (please provide the reference)

3. Please specify whether the assessment of left atrial deformation was performed based on algorithms prepared for assessing left atrial deformation or whether the software originally intended to assess the left ventricle was adopted.

4. Please describe the transplantation technique in the “Patient characteristic” paragraph.

5. I propose to rewrite paragraph 2.4 (Strain analysis) in such a way as to separate the strain and strain rate parameters. In this version, It is difficult to distinguish between the strain and strain rate parameters.

6. The authors stated that the frame rates were 61-95 fps. As the strain rate is a time-dependent parameter, it requires high frame rates (>100 fps) even at rest (see: J-U Voigt et al. “Definitions for a common standard for 2D speckle tracking echocardiography: consensus document of the EACVI/ASE/Industry Task Force to standardize deformation imaging”, https://doi.org/10.1093/ehjci/jeu184). Please comment.

Sincerely,

Round 2

Reviewer 1 Report

I thank the authors for their revision. I acknowledge that the study was carried out with great efforts and expertise of the authors. Yet, despite being a hypothesis-generating observational study, I still believe that the background rationale for choosing the study aim should be presented. Also, the inferences caused by the small sample size with large deviations of the observed results yield limited clinical significance.

Thank you.
